# NUMSnet: Nested-U Multi-Class Segmentation Network for 3D Medical Image Stacks

**Sohini Roychowdhury** [1,2]

1   AI Engineering, Accenture LLP, Palo Alto, CA 94304, USA; sohini.roychowdhury@accenture.com
2   Adjunct Faculty, Computer Engineering, Santa Clara University, Santa Clara, CA 95053, USA

**Abstract:** The semantic segmentation of 3D medical image stacks enables accurate volumetric reconstructions, computer-aided diagnostics and follow-up treatment planning. In this work, we present a novel variant of the Unet model, called the *NUMSnet,* that transmits pixel neighborhood features across scans through nested layers to achieve accurate multi-class semantic segmentation with minimal training data. We analyzed the semantic segmentation performance of the NUMSnet model in comparison with several Unet model variants in the segmentation of 3–7 regions of interest using only 5–10% of images for training per Lung-CT and Heart-CT volumetric image stack. The proposed NUMSnet model achieves up to 20% improvement in segmentation recall, with 2–9% improvement in *Dice* scores for Lung-CT stacks and 2.5–16% improvement in *Dice* scores for Heart-CT stacks when compared to the Unet++ model. The NUMSnet model needs to be trained with ordered images around the central scan of each volumetric stack. The propagation of image feature information from the six nested layers of the Unet++ model are found to have better computation and segmentation performance than the propagation of fewer hidden layers or all ten up-sampling layers in a Unet++ model. The NUMSnet model achieves comparable segmentation performance to previous works while being trained on as few as 5–10% of the images from 3D stacks. In addition, transfer learning allows faster convergence of the NUMSnet model for multi-class semantic segmentation from pathology in Lung-CT images to cardiac segmentation in Heart-CT stacks. Thus, the proposed model can standardize multi-class semantic segmentation for a variety of volumetric image stacks with a minimal training dataset. This can significantly reduce the cost, time and inter-observer variability associated with computer-aided detection and treatment.

**Keywords:** semantic segmentation; multi-class; 3D image stacks; region of interest; Dice score; Unet; CT images; overfitting

## 1. Introduction

Deep learning approaches for vision-based detection have seen significant breakthroughs over the past five years [1]. From autonomous driving to virtual reality and from facial detection for phone unlocking to home security camera systems, several deep-learning-based object detection and segmentation models have been developed to date to keep up with speed, precision and hardware requirements [2]. For semantic segmentation tasks, where objects or regions of interest (ROIs) are enclosed within a closed polygon, the Unet model [3] and its variants have been a widely preferred method owing to the relatively low computational complexity and high adaptability across use cases due to short- and long-range skip connections. This allows the Unet and variant models to be well trained from only a few hundred images, as opposed to requiring thousands of annotated images for deep learning models with dense connections, which are preferred in the real-time use cases of autonomous driving and augmented reality [4,5]. However, segmenting multiple ROIs with varying sizes and shapes in continuous image stacks or videos can be challenging due to the biases introduced by foreground regions with varying sizes and can result in jittery detection across subsequent frames. In this work, we present a novel

Unet model variant, called the NUMSnet, that is capable of accurately segmenting multiple shapes and sizes of ROIs and transferring information across subsequent images in a stack to provide superior segmentation performance while training with only a fraction of the training images when compared to state-of-the art approaches.

Deep learning approaches for medical imaging use cases have a unique requirement to maintain high recall for pathological detection: i.e., the over-detection of pathology is acceptable since a specialist will always look at the report and confirm, but detection failures must be minimized. To enhance the *quality of detection* and enable explainability, semantic segmentation and explainable classification models are preferred across medical domain use cases to identify pathology in patients and also localize the pathological sites. For example, the recent work in [6] demonstrated the significance of segmentation for detecting Leukemia using bloodstream images. A review of several deep learning approaches/models developed to detect pathology is presented in [7] for the use case of detecting monkeypox from RGB images. Another work in [8] reviewed recent advances in deep learning models for chest disease detection using X-ray images. Unet and its variant models continue to be the preferred method in the medical imaging domain owing to the fewer parameters involved when compared to dense-connection models. The Unet and variant models have thus far been applied to a variety of medical imaging use cases, from dental segmentation in [9] and human skin classification in [10] to polyp segmentation tasks in colonoscopy images in [11]. This demonstrates the versatility of the Unet and variant models in the medical imaging domain, thereby necessitating the development of advanced Unet model versions.

The medical imaging domain often requires the semantic segmentation of multiple ROIs, also known as multi-class segmentation, from 3D medical image stacks of CT or MRI images for diagnostics and pre-procedural planning tasks. Performing such segmentation manually can be both costly and time-intensive [12]. Additionally, the manual segmentation process suffers from inter-observer variability, where two medical practitioners may disagree on the exact locations of the ROIs [12]. The challenge associated with training deep learning models using annotated images isolated across patient stacks is that 3D medical image stacks often have variable pixel resolutions and vary in additive image noise owing to imaging and storage conditions. This can impede the scalability of automated deep learning solutions to other patient image stacks acquired under different imaging settings [13]. Several previous works on medical image semantic segmentation performed binary segmentation for each image [3,14] or two-stage multi-class segmentation for image stacks [15] to overcome such image-level variations. In this work, we present a novel single-stage variant of the Unet model, as shown in Figure 1, that propagates image features across scans, which results in faster network convergence with few training images for volumetric medical image stacks. The proposed NUMSnet model requires training images to be in order, but not necessarily subsequent images in a sequence. The training set is shown in Figure 1 for the image stack $[T_0$ to $T_n]$. Once trained, the test set of images can be ordered or randomized per stack, as represented by sets $S_m$ and $S_{m'}$ in Figure 1.

The novel multi-class semantic segmentation NUMSnet model presented in this paper achieves multi-class semantic segmentation with only 10% of frames per 3D image stack. It is noteworthy that the proposed model has significantly less computational complexity than the 3D Unet model and its variants that perform 3D convolutions across image stacks but has a comparable volumetric segmentation performance [16]. We investigated three main analytical questions regarding the multi-class semantic segmentation of 3D medical image stacks. (1) Does the transmission of image features from some of the layers of a Unet variant model enhance the semantic segmentation performance for multi-class segmentation tasks? (2) Is the order of training and test frames significant to segmentation tasks for 3D volumes? (3) How many layers should be optimally propagated to ensure model optimality while working with sparse training data? The key contributions of this work are as follows:

1. A novel multi-scan semantic segmentation model that propagates feature-level information from a few nested layers across ordered scans to enable feature learning from as few as 10% of annotated images per 3D medical image stack.
2. The transfer learning performance analysis of the proposed model compared to existing Unet variants on multiple CT image stacks from Lung-CT (thoracic region) scans to Heart-CT regions. The NUMSnet model achieves up to 20% improvement in segmentation recall and 2–16% improvement in *Dice* scores for multi-class semantic segmentation across image stacks.
3. The identification of a minimal number of optimally located training images per volumetric stack for multi-class semantic segmentation.
4. The identification of the optimal number of layers that can be transmitted across scans to prevent model over- or underfitting for the segmentation of up to seven ROIs with variables shapes and sizes.

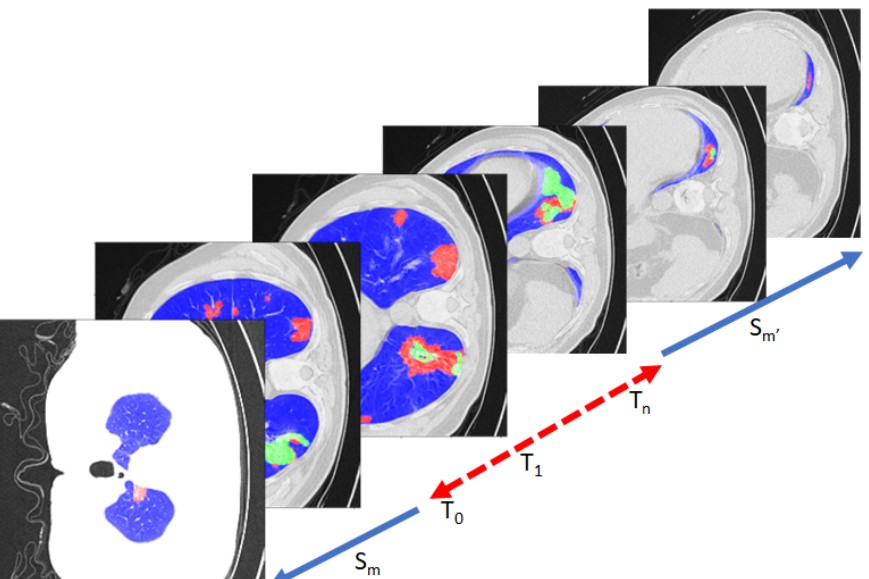

**Figure 1.** An example of the proposed NUMSnet system on a Lung-CT image stack. The training images (*T*) are selected in order or in sequence, while test images (*S*) can be random or in sequence.

This paper is organized as follows. The existing literature and related works are reviewed in Section 2. The datasets under analysis and the NUMSnet model are explained in Section 3. The experiments and results are shown in Section 4. A discussion regarding the limiting conditions is presented in Section 5, and the final conclusions are presented in Section 6.

## 2. Related Work

Deep learning models have been highly popular for computer-aided detection in the past decade and preferred over the signal processing methods in [17,18]. This is primarily due to the ability of deep learning models to automatically learn features that are indicative of an ROI if a significant volume of annotated data is provided. Signal processing models, on the other hand, rely on hand-generated features that may lead to faulty detections due to the high variability across imaging modalities and storage and transmission formats. The prior work in [19] demonstrates a two-path CNN model that can take filtered Lung-CT images followed by fuzzy c-means clustering to segment the opacity in each Lung-CT image. While such feature-based works have low data dependence, the models often do not scale across datasets.

Unet models with the default 2D architecture have been used extensively for medical image segmentation applications since 2015 [3]. While other deep learning models, such as MaskRCNN [20] and fully convolutional neural networks (FCNs) [21], are more popular

in non-medical domains, Unet and its variants have continued to be the preferred deep learning model for medical image segmentation tasks. The 2D Unet model and its variants apply long and short skip connections that ensure that the number of trainable parameters is low, thereby leading to quicker training with fewer images. Over the last few years, several Unet model variants have been applied for dense volumetric scan segmentation. In instances where high volumes of annotated data are readily available, such as anatomical regions in Heart-CT scans in [15], multi-stage Unet variants have been introduced. The works in [15,22] trained two separate Unet models with separate loss functions, with the objective of zooming into the foreground regions in the first network, followed by separating the foreground into various ROIs. Other variants of multi-2D-Unet models, such as the work in [23], implement trained Unet models at different resolutions, i.e., one Unet model trained on images with dimensions of [256 × 256], another trained at a resolution of [512 × 512] and so on for lung segmentation. However, these methods require significantly high volumes of annotated data to train multiple Unet models.

Other recent works in [13,24] applied variations to the 2D Unet model to achieve the segmentation of opacity and lung regions in chest CT scans to aid in COVID-19 detection. Additionally, in [25], Inf-net and Semi-Inf net models are presented that can perform binary segmentation for lung opacity detection with *Dice* scores in the range of 0.74–0.76. Most of these existing methods require several hundred annotated training images across scans and patients and can efficiently be trained for binary semantic segmentation tasks.

Some of the well-known 2D Unet model variants used in the medical imaging domain are the wide Unet (wU-net) and Nested Unet (Unet++) [14]. While a typical Unet model with a depth of 5 will have filter kernel widths of [32, 64, 128, 256, 512] at model depths of 1 through 5, the wUnet model has filter kernel widths of [35, 70, 140, 280, 560] at model depths of 1 through 5. Thus, wUnet has more parameters and thus can enhance segmentation performance when compared to Unet. The Unet++ model, on the other hand, generates dense connections with nested up-sampling layers to further enhance the performance of semantic segmentation, as presented in [26,27]. In this work, we propose an enhanced Unet++ architecture called the NUMSnet, where the features from the nested up-sampling layers are transmitted across scans for increased attention to smaller regions of interest (such as opacity in Lung-CT images). This layer propagation across scans enables multi-class semantic segmentation with only 10% of annotated images per 3D volume stack.

Another major family of Unet models that have been applied to volumetric image segmentation tasks in the medical imaging domain is the 3D Unet model variants, as shown in [16]. These models have significantly higher computational complexity when compared to the 2D Unet model and its variants due to the 3D convolutions in each layer, but they achieve superior segmentation performance for pathological sites in 3D image stacks. Another work in [28] combined the Resnet backbone with the 3D Unet model to improve the resolution of segmentation for small ROIs in Lung-CT images. As additional variants of the 3D Unet model, the encoder architecture can be modified with the VGG19, 3D ResNet152 or DenseNet201 backbone to achieve 80–98% *Dice* scores for multi-class semantic segmentation tasks [16]. However, these 3D Unet models are difficult to train and may need around 1700 epochs and 1.8 h to train on single-GPU systems. Besides 3D Unets, another recent work that implemented a 3D fully convolutional neural network model for volumetric segmentation is shown in [29], where MRI stacks are segmented with about an 86% *Dice* score with over 48 h of training time. It is noteworthy that 3D Unet models have been preferred for cardiac CT segmentation so far, with the work in [15] applying a two-stage 3D Unet model for voxel-level segmentation of the heart. Another work in [30] implemented a deeply supervised 3D Unet model with a multi-branch residual network and deep feature fusion along with focal loss to achieve 86–96% *Dice* scores for the semantic segmentation of small and large ROIs. Our work aimed to perform 2D semantic segmentation and achieve a comparable segmentation performance to 3D model variants with under 10 min of training time on a single GPU system.

## 3. Materials and Methods

### 3.1. Data: Lung-CT and Heart-CT Stacks

In this work, we analyze two kinds of single-plane volumetric CT image stacks. In the first category, Lung-CT image stacks were collected from the Italian Society of Medical and Interventional Radiology. The first Lung-CT (Lung-med) volumetric stack [31] contains 829 images from a single 3D image stack with [512 × 512]-dimension images. Out of these 829 scans, 373 are annotated. The second dataset (Lung-rad) contains 9 axial volume chest CT scans with 39–418 images per stack. All Lung-CT images are annotated for 3 ROIs, namely, ground-glass opacity (GGO), consolidations and the lung region as the foreground and can be downloaded from [32].

In the second category, the Heart-CT image dataset is from the MICCAI 2017 Multi-Modality Whole Heart Segmentation (MM-WHS) challenge [15,30], from which we selected the first 10 training CT image stacks of the heart region for analysis. This dataset contains coronal volumetric stacks with 116–358 images per volume and multi-class semantic segmentation annotations for up to 7 heart-specific ROIs represented by label-pixel values of [205, 420, 500, 550, 600, 820, 850], respectively. These pixel regions represent the left ventricle blood cavity (LV), myocardium of the left ventricle (Myo), right ventricle blood cavity (RV), left atrium blood cavity (LA), right atrium blood cavity (RA), ascending aorta (AA) and pulmonary artery (PA), respectively. It is noteworthy that for the Heart-CT dataset, only 10–15% of the images per stack contain annotated ROIs. Thus, when selecting the ordered training dataset, it was ensured that at least 50% of the training samples contained annotations. Some examples of the Lung-CT and Heart-CT images and their respective annotations are shown in Figure 2.

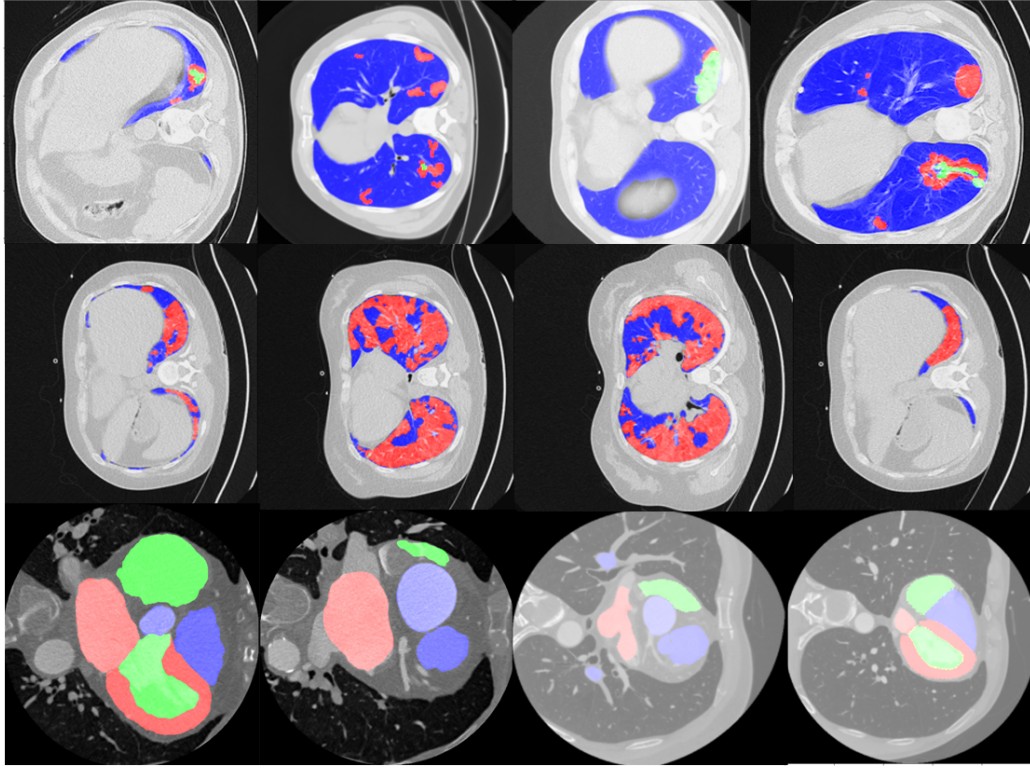

**Figure 2.** Examples of multi-class segmentation datasets used in this work. **Row 1**: Lung-med dataset. **Row 2**: Lung-rad dataset. For **Row 1** and **Row 2** the regional color coding is as follows. Blue: lung region; Red: GGO; Green: consolidation regions. **Row 3**: Heart-CT dataset. The ROIs are color-coded as follows. Red plane: label pixels 205 and 420. Blue plane: label pixels 500 and 550. Green plane: label pixels 600, 820 and 850.

### 3.2. Image Data Pre-Processing

Each image from the data stacks under analysis here was pre-processed for the Unet and variant models. First, each input image was resized to [256 × 256 × 1] for ease of processing. Next, the resized image $I$ was re-scaled to the range [0,1], thereby resulting in image $I'$, using min–max normalization, as shown in (1), where $min_I$ and $max_I$ refer to the minimum and maximum pixel values in $I$. This is followed by the generation of multi-dimensional label vectors [256 × 256 × $d$] per image, where $d$ represents the number of classes that each pixel can be classified into. These label vectors are generated as binary images for each class. For example, the Heart-CT stack images contain up to 7 different annotated regions depicted by a certain pixel value $pix_i, \forall i = [1:7]$. Thus, the ground-truth label vector ($G'$) generated per image contains 7 planes, where each plane $G'_i$ is generated as a binary mask from the label masks ($G$), as shown in (2). This process defines the ground-truth G' such that the Unet decision-making function ($f_i$) proceeds to analyze whether each pixel belongs to a particular class $i$ or not. Finally, the output is a $d$-dimensional binary image ($P$), where each image plane ($P_i$) is thresholded at a pixel value $\tau = 0.5$, as shown in (3).

$$I' = \frac{I - min_I}{max_I - min_I}. \tag{1}$$

$$\forall i \in [1:d], G'_i = [G == pix_i], \tag{2}$$

$$and, P_i = [f_i(I') > \tau]. \tag{3}$$

Once the datasets are pre-processed, the next step is to separate the data stacks into training, validation and test sets. There are two ways in which the training/validation/test datasets are sampled for each volume stack. The first is the random sampling method, where 10% of the scans per volume are randomly selected in ascending order for training, 1% of the remaining images are randomly selected for validation, and all remaining images are used for testing. The second is the sequential sampling method, which starts from a reference scan in the volumetric stack. This reference scan can either be the first or middle scan in the stack. We sample 10% of the total number of images in the stack starting from the reference scan in sequence, and these become the training set of images. From the remaining images, 1% can be randomly selected for validation, while all remaining scans are test set images in sequence. Using these methods, we generated training sets with the sizes [82 × 256 × 256 × 1], [84 × 256 × 256 × 1] and [363 × 256 × 256 × 1] for the Lung-med, Lung-rad and Heart-CT stacks, respectively.

### 3.3. Unet Model Variant Model Implementation

To date, Unet and its variants, such as wUnet and Unet++ models, have been applied to improve foreground segmentation precision for small ROIs, as shown in [14,30]. One major difference between the Unet++ and Unet models [3] is the presence of nested layers that combine the convolved and pooled layers with the up-sampling (transposed convolutional) layers at the same level. Thus, for a Unet with a depth of 4, a Unet++ model results in 6 additional nested layers, shown as [X(1,2), X(1,3) X(1,4), X(2,2), X(2,3), X(3,2)] in Figure 3. These additional layers increase the signal strength at each depth level and amplify the segmentation decisions around boundary regions of ROIs [14]. We selected an optimal depth of 4 for our analysis of Unet and variant models based on the prior work in [33], which showed superior semantic segmentation at a depth of 4 when compared to shallower Unet models. Additionally, depth-4 Unet and variant models are preferred for an appropriate comparative analysis with previous works.

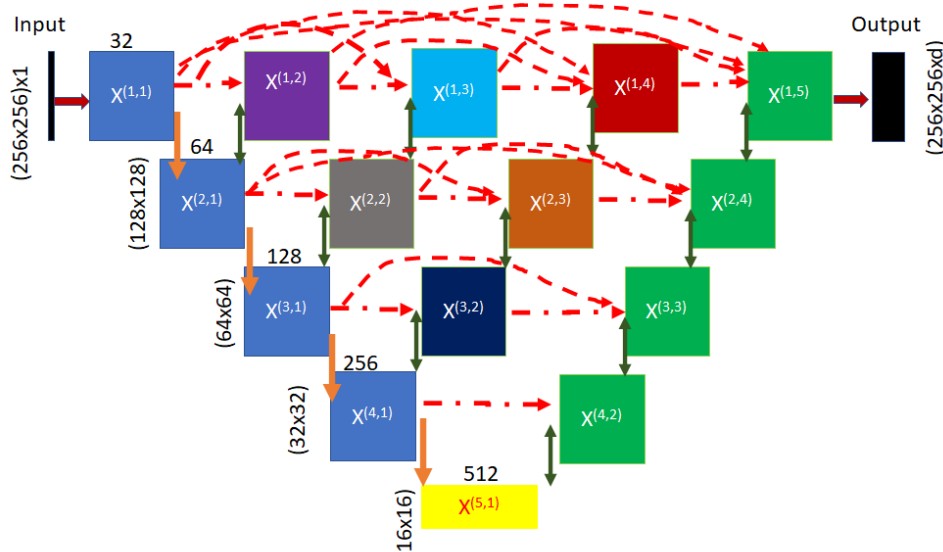

**Figure 3.** Example of a Unet++ model with a depth of 4. The global feature layer is X(5,1), and the depth is associated with the distance of each layer from the global feature layer. The blue layers correspond to convolved and pooled layers. The green layers correspond to merged transposed convolutions, followed by convolution outcomes from the same depth layers. The 6 additional nested color-coded layers (purple, cyan, red, gray, orange, dark and blue, corresponding to [X(1,2), X(1,3), X(1,4), X(2,2), X(2,3), X(3,2)], respectively) contain spatial pixel neighborhood information, that can be transmitted temporally across images/scans for an increased accuracy of semantic segmentation.

A Unet model comprises encoder and decoder layers, where the encoder layers perform convolution followed by a max-pooling operation, while the decoder layers perform concatenation followed by up-sampling and convolution operations. Starting with the input image $I'$, the encoder layers are [X(1,1), X(2,1), X(3,1), X(4,1)], respectively. The output of each encoder layer results in an image with half the input dimensions but with additional feature planes. For example, the input to layer X(1,1) is an image with the size [256 × 256 × 1], while the output has dimensions of [128 × 128 × 32] due to convolution with a [3 × 3] kernel with a width of 32 and max-pooling with a [2 × 2] kernel. Thus, at the final level (X(5,1)), a global feature vector with the size [16 × 16 × 512] is generated. At this point, the decoder layers [X(4,2), X(3,3), X(2,4), X(1,5)] convert the dense features back to the segmented image planes. The decoder layers concatenate the up-sampled features with the encoder layer outputs from the same level to promote a better distinction between foreground pixels (scaled value of 1) versus background pixels (scaled value of 0). For example, at depth level 1 from the global feature layer, the output from layer X(4,1) is concatenated with the up-sampled image from layer (5,1), resulting in image features with dimensions of [32 × 32 × 512] that are then subjected to convolutions in layer (4,2), thereby resulting in image features with dimensions of [32 × 32 × 256].

The Unet++ model, on the other hand, was developed to enhance the boundary regions for relatively small ROIs by introducing nested decoder layers at each depth level, as shown in Figure 3. The 6 additional nested/hidden decoder layers that are introduced in the skip connection pathway are [X(1,2), X(1,3), X(1,4), X(2,2), X(2,3), X(3,2)] in Figure 3. The 2D weights for each decoder layer X in image $n$ with encoder layer index $i'$ and decoder layer index $j'$ (i.e., $x^n(i', j')$) are generated using Equation (4), as shown in [14]. Here, $\zeta(.)$ refers to the convolution operation, $v(.)$ refers to the up-sampling operation and [.] refers to concatenation.

$$x^n(i', j') = \begin{cases} \zeta(x^n(i'-1, j')) & j' = 1 \\ \zeta([[x^n(i', k')]_{k'=1}^{j'-1}], v(x^n(i'+1, j'-1))]) & j' > 1. \end{cases} \tag{4}$$

For example, the decoder layer outcome $x^n(4,2) = \zeta[[x^n(4,1), v(5,1)]]$ using Equation (4). This ensures additional skip connections that lead to improved region boundary detection.

The primary parameters that need to be tuned to ensure the optimal training of the Unet or variant model are the following: data augmentation methods, batch size, loss function, learning rate and reported metric per epoch. In this work, we applied image data augmentation using the tensorflow keras library by augmenting images randomly to ensure a rotation range, width shift range, height shift range and shear range of 0.2 and a zoom range of [0.8, 1] per image. Since the training dataset has few samples, we implemented a training batch size of 5 for the Lung-CT images and a batch size of 10 for Heart-CT images. It is noteworthy that the batch size should scale with the number of detection classes; thus, we used additional images per batch for the Heart-CT stack. For all Unet and variant models, we used the Adam optimizer with a learning rate of $10^{-4}$. Finally, the metrics under analysis are shown in (5)–(8) based on the work in [34]. For each image with $l$ pixels and $d$ image planes for the ground-truth ($G_i'$), the intersection over union ($IoU$) or Jaccard metric in (5) represents the average fraction of correctly identified ROI pixels. Precision ($Pr$) in (6) and recall ($Re$) in (7) denote the average fraction of correctly detected ROI pixels per predicted image and per ground-truth image plane, respectively. The *Dice* coefficient in (8) further amplifies the fraction of correctly classified foreground pixels. The *Dice* coefficient can also be derived from the precision ($Pr_i$) and recall ($Re_i$) metrics per image plane, as shown in (8).

$$IoU = \sum_{i=1}^{d} \sum_{j=1}^{l} \frac{|P_i(j) \cap G_i'(j)|}{P_i \cup G_i'}, \tag{5}$$

$$Pr = \sum_{i=1}^{d} \sum_{j=1}^{l} \frac{P_i(j) \cap G_i'(j)}{P_i(j)}, \tag{6}$$

$$Re = \sum_{i=1}^{d} \sum_{j=1}^{l} \frac{P_i(j) \cap G_i'(j)}{G_i'(j)}, \tag{7}$$

$$Dice = \sum_{i=1}^{d} \sum_{j=1}^{l} \frac{2 * |P_i(j) \cap G_i'(j) + 1|}{P_i(j) + G_i'(j) + 1} = \sum_{i=1}^{d} \frac{2 * Pr_i * Re_i}{Pr_i + Re_i}. \tag{8}$$

The loss functions under analysis are shown in (9)–(11). The *Dice* coefficient loss ($DL$) in (9) is the inverse of the *Dice* coefficient, so it ensures that the average fraction of correctly detected foreground regions increases for each epoch. The binary cross-entropy loss ($BCL$) in (10) is a standard entropy-based measure that decreases as the predictions and ground-truth become more alike. Finally, the binary cross-entropy-Dice loss ($BDL$) in (11) is a combination of BCL and DL based on the work in [14].

$$DL = -D, \tag{9}$$

$$BCL = -\sum_{i=1}^{d} \sum_{j=1}^{l} [P_i(j) log(G_i'(j))], \tag{10}$$

$$BDL = \frac{BCL}{2} + DL. \tag{11}$$

Finally, we analyze the loss function curves per epoch using the deep-supervision feature from the Unet++ model [14] in Figure 4. Here, we assessed convergence rates for outputs at each depth level. In Figure 4, we observe that the curves for the convergence of

outputs from depths 4 and 3 (i.e., layers X(1,5) and the resized output of X(2,4)) are relatively similar and better than the loss curves at depth 1 (resized output of layer X(4,2)). This implies that as the transposed convolutions move farther away from the global feature layer X(5,1), additional local feature-level information gets added to the semantic segmentation output. Thus, for a *well-trained* Unet++ model, the initial transposed convolution layers closer to the global feature layer X(5,1) add less value to the semantic segmentation task when compared to the layers farther away from it. This variation in loss curves at the different depth levels, based on the work in [35], demonstrates the importance of the additional nested up-sampling layers to the final multi-class segmented image.

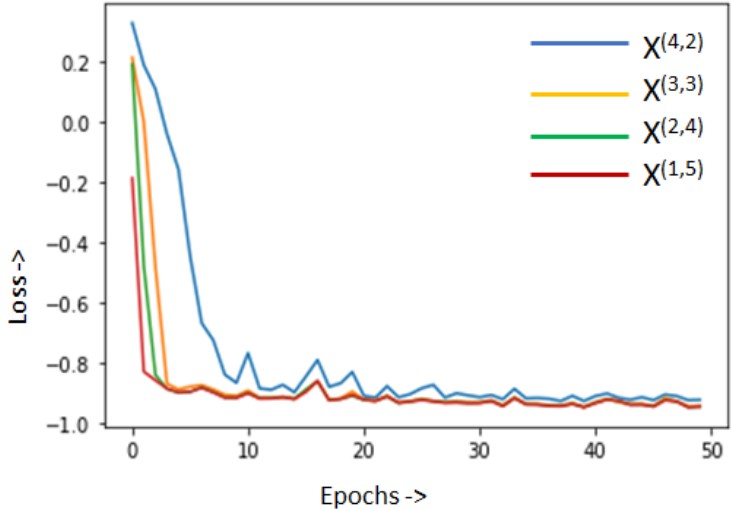

**Figure 4.** Example of loss functions per depth layer in Unet++ model using the deep-supervision feature on the Lung-med training dataset. The resized image outcome from X(4,2) achieves lower segmentation resolution when compared to the outcome from X(1,5). Thus, nested layers enhance local boundary-region-specific features for segmentation.

### 3.4. The NUMSnet Model

While the Unet and variant models are efficient in the 2D segmentation of each scan, segmenting volume stacks requires further intervention wherein pixel neighborhood information can be transmitted to the next ordered scan, thereby allowing better resolution of semantic segmentation while training on few images. The NUMSnet model is a 3D extension of the Unet++ model, wherein the outcomes of the nested Unet++ layers are transmitted to subsequent scans. As a first step for an image $n$, the 2D weights for the decoder layers are computed using Equation (4). Next, the 2D weights of the nested layers are transmitted using Equation (12), where the final 3D weight for each hidden layer $(X^n(i', j'))$ is computed by concatenating the weights of the same layer $(i', j')$ from the previous scan, followed by the convolution operation. For the first image in each stack, the hidden layers are concatenated with themselves, followed by convolution, as shown in Equation (12).

$$X^n(i', j') = \begin{cases} \zeta([x^n(i', j'), x^n(i', j')]) & n = 1 \\ \zeta([x^{n-1}(i', j'), x^n(i', j')]) & n > 1, \forall i <= j. \end{cases} \tag{12}$$

From the implementation perspective, for the NUMSnet model, we applied batch normalization to encoder layers only and dropout at layers X(4,1) and X(5,1) only (GitHub code available at https://github.com/sohiniroych/NUMSnet, accessed on 12 June 2023). In addition, the widths of kernels per depth layer for the NUMSnet model are [5, 70, 140, 280, 560], similar to those of the wUnet model. This process of transmitting and concatenating layer-specific features with those of the subsequent ordered images generates finer boundaries for ROIs. This variation in the Unet++ model to generate the NUMSnet

model is shown in Figure 5. The additional layers generated in this process are shown in the model diagrams in Appendix A Figure A1.

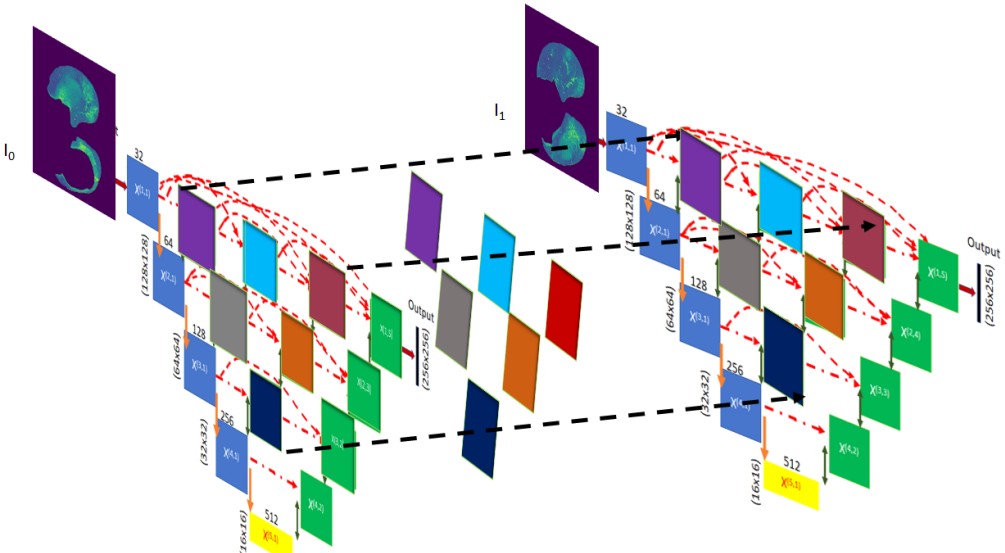

**Figure 5.** The proposed NUMSnet, which propagates the image features from the 6 nested layers across scans. The outcome of each nested layer is concatenated and convolved with the equivalent layer of the subsequent ordered image in the 3D stack.

The NUMSnet model has two key hyper-parameters. First, the relative location of the training scans in the 3D volume stack impacts the training phase. Since layer information is transmitted to the subsequent ordered scans, selecting training scans that contain ROIs in several subsequent scans is important. We analyze this sensitivity to the training data location in a 3D stack by varying the location of the reference training frame from the beginning to the middle of the stack, followed by selecting the subsequent or randomly selected frames in order. For example, this ensures that in the Heart-CT stacks, if an aortic region is detected for the first time in a scan, the ROI first increases and then decreases in size as training progresses. The second hyper-parameter for the NUMSnet model is the number of decoder layers that can be transmitted across scans. If all 10 decoder layers [X(1,2), X(1,3), X(1,4), X(1,5), X(2,2), X(2,3), X(2,4), X(3,2), X(3,3), X(4,2)] in Figure 3 are transmitted to the subsequent scans, this would incur high computational complexity (14.5 million trainable parameters). Thus, we analyze the segmentation performance using this NUMSnet variant (called NUMS-all), where features from all 10 up-sampling layers are transmitted. The primary reason for transmitting only up-sampling layers is that up-sampling generates image feature expansion based on pixel neighborhood estimates. Thus, information added during the up-sampling process further aids in the foreground versus background decision-making process per image plane.

## 4. Experiments and Results

In this work, we analyze the performance of the Unet model and its variants for the multi-class semantic segmentation of volumetric scans using only 10% of the annotated data for training. To analyze the importance of nested layer propagation across subsequent images, we performed five sets of experiments. First, we comparatively analyze the segmentation performance per ROI for the NUMSnet when compared to the Unet [3] model and its variants [14] for the Lung-CT image stacks. Second, we analyze the sensitivity of the NUMSnet model to the relative position and selection of training data for randomly ordered sampling versus sequential sampling from the beginning or middle of the volumetric stack. Third, we analyze the semantic segmentation performance of the NUMSnet model when only nested layer features are transmitted versus when all up-sampling layer features are transmitted (NUMS-all). Fourth, we assessed the semantic segmentation capability of the

NUMSnet in comparison with Unet variants for the transfer learning of weights and biases from segmenting three ROIs (in Lung-CT stacks) to segmenting seven ROIs (in Heart-CT stacks). Finally, we performed an ablation study, in which we assessed the importance of each hidden layer to the superior semantic segmentation performance of the NUMSnet model. We compared the segmentation performance when the selected hidden layers per level are propagated. We also comparatively analyze the performance of the NUMSnet with respect to state-of-the-art models that were trained on higher volumes of data.

It is noteworthy that while the training samples are ordered, the test samples may be out of order, starting at the other end of the stack or starting at a new volumetric stack. In the testing phase, the nested layer outputs and model layer weights and biases are collected per test image and passed to the next image. Once the NUMSnet model is optimally trained, the out-of-order scans in the test stacks do not significantly impact the segmentation outcomes. All other parameters, including data augmentation, loss functions, batch size, compiler, learning rate and reported metrics, are kept similar to those of the Unet model and variants to realize the segmentation enhancements per epoch.

An additional consideration for segmenting medical images is that relative variations in pixel neighborhoods are significantly less than those in regular camera-acquired images, such as those used for autonomous driving or satellite imagery [4]. Thus, feature-level propagation across scans through the NUMSnet model enhances the decision making around boundary regions, especially for smaller ROIs. However, the additional nested layer transmission introduces a higher number of parameters in the Unet variant models, which leads to a slower training time and higher GPU memory requirements for model training. In this work, we used Nvidia RTX 3070 with 8GB of GPU RAM on an Ubuntu Laptop and tensorflow/keras libraries to train and test the volume segmentation performance. In instances where models have a high number of parameters, keeping a small batch size of 5–10 ensures optimal model training. We collectively analyzed the segmentation performance along with the computational complexities incurred by each model to demonstrate the ease of use and generalization to new datasets and use cases.

### 4.1. Multi-Class Segmentation Performance of Unet Variants

For any multi-class semantic segmentation model, it is important to assess the computational complexity introduced by additional layers in terms of the number of trainable parameters jointly with the semantic segmentation performance. Table 1 shows the variations in the number of trainable and non-trainable parameters for all 2D Unet variants analyzed in this work. Here, we found that Unet is the fastest model, while NUMS-all has almost twice the number of trainable parameters when compared to Unet. In addition, the NUMSnet model is preferable to NUMS-all with regard to computational complexity, as it has less of a chance of overfitting [36]. Since the NUMSnet model performs 2D operations in encoder and decoder layers, we comparatively analyzed its performance with the 2D model variants only.

**Table 1.** Variations in the number of parameters in Unet model variants.

| Model | Total Params | Trainable Params | Non-Trainable Params |
|---|---|---|---|
| Unet | 7,767,523 | 7,763,555 | 3968 |
| wUnet | 9,290,998 | 9,286,658 | 4340 |
| Unet++ | 9,045,507 | 9,043,587 | 1920 |
| NUMSnet | 11,713,943 | 11,711,843 | 2100 |
| NUMS-all | 14,526,368 | 14,524,268 | 2100 |

It is noteworthy that the base 3D Unet model, as shown in [16,28], has 19,069,955 total parameters, which increases rapidly with modifications to the encoder–decoder block backbones. Next, we analyze the multi-class semantic segmentation performance of the NUMSnet and Unet model variants. In Table 2, the average semantic segmentation across

five randomly ordered training dataset selections of the three ROIs in the Lung-med dataset is presented. Here, we observe that the performance of lung segmentation is the best and similar across all Unet variants, with a *Dice* score ranging between 92 and 96%. This is intuitive since the lung is the largest region that is annotated in most images. The Unet and variant models preferentially extract this ROI with minimal training data. We also observe that for the segmentation of opacity (GGO) and consolidation (Con) regions, the NUMSnet model has the best combination of *Pr* and *Re*, thereby resulting in 2–8% higher *Dice* scores than all Unet variants. The Unet++ model, on the other hand, achieves superior overall *Pr* metrics but low *Re* metrics, which leads to lower *Dice* and *IoU* scores. Some examples of Unet and variant model segmentation are shown in Figure 6. Here, we observe that for small as well as large ROIs, the NUMSnet has better segmentation resolution when compared to all other Unet variants.

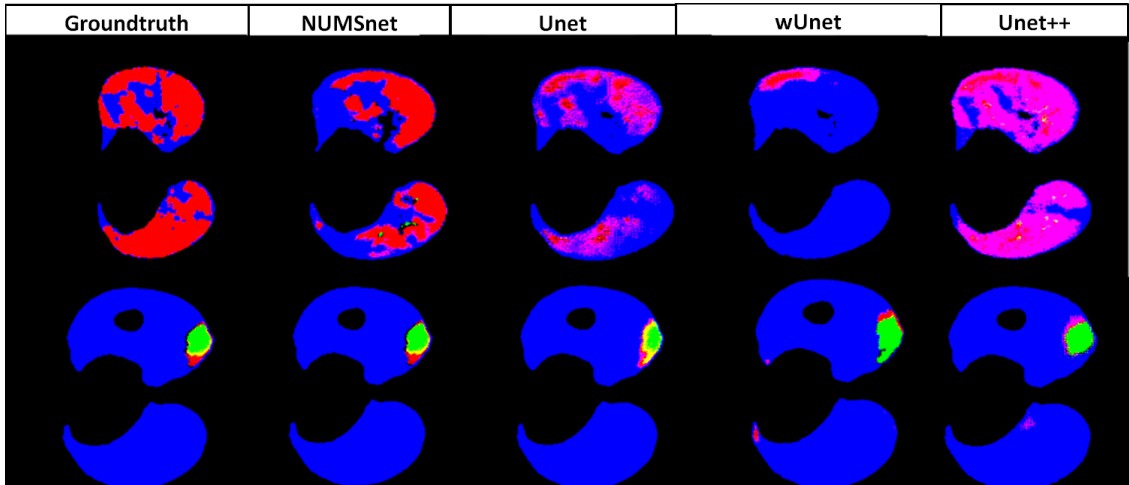

**Figure 6.** Example of Lung-CT segmentation by the Unet variant models. **Row 1** represents poor segmentation results. **Row 2** represent good segmentation results since the major ROI is the lung. The color coding is as follows. Blue: lung regions; Red: GGO regions; Green: consolidation regions; Magenta: over-detection of consolidation regions.

For all Unet variants under analysis, the number of epochs is 60, and the optimal loss function is the BDL with the *Dice* coefficient as the reported metric. We observe poor convergence with the DL loss function since the large lung regions are weighted more by the DL, thereby resulting in the high accuracy of lung segmentation but poor performance for GGO and consolidation segmentation.

Next, we analyze the segmentation performance on smaller Lung-CT stacks from radiopedia (Lung-rad), and the results are shown in Table 3. For the lung region segmentation, we have similar observations on this dataset to those on the Lung-med dataset. All Unet variants models achieve 95–96% *Dice* scores for the segmentation of the large lung region. However, for segmenting GGO and Con regions, the NUMSnet model achieves higher *Re* and up to 10% improvement in *Dice* coefficients over the other Unet variant models. Examples of good and bad selected segmentation on this dataset are shown in Figure 7. Here, we observe that the lung region is well detected by all Unet model variants, but Unet misclassifies the GGO as consol (in row 2, red regions are predicted as green), while the NUMSnet under-predicts the GGO regions. The reason for the lower performance for the Lung-rad stacks when compared to the Lung-med stack is that the number of frames in the sequence for training per stack is lower when compared to the Lung-med stack. Thus, for denser volumetric stacks, the NUMSnet has better multi-class segmentation performance when compared to shorter stacks with few images.

**Table 2.** Comparative performance of Unet and variant models on the Lung-med stack averaged over 5 runs. The best values for each metric are highlighted.

| Task | *Pr* | *Re* | *IoU* | *Dice* |
|---|---|---|---|---|
| NUMSnet, Con | 82.06 | 65.86 | **57.43** | **61.25** |
| NUMSnet, GGO | 89.86 | 85.87 | **78.76** | **81.29** |
| NUMSnet, Lung | 97.35 | **94.96** | **92.94** | **95.9** |
| Unet, Con | **91.91** | 32.48 | 30.43 | 33.84 |
| Unet, GGO | 90.56 | 73.69 | 68.26 | 70.92 |
| Unet, Lung | 91.66 | 94.31 | 86.59 | 92.2 |
| wUnet, Con | 64.02 | **77.85** | 53.42 | 53.66 |
| wUnet, GGO | 81.92 | **95.29** | 78.33 | 80.43 |
| wUnet, Lung | 99.27 | 91.47 | 90.94 | 94.35 |
| Unet++, Con | 71.67 | 57.14 | 42.21 | 45.36 |
| Unet++, GGO | **92.87** | 71.54 | 68.06 | 71.18 |
| Unet++, Lung | **99.61** | 90.41 | 90.17 | 93.89 |

**Table 3.** Averaged performance of Unet and variant models on 10 Lung-rad CT stacks across 5 runs. The best values for each metric are highlighted.

| Task | *Pr* | *Re* | *IoU* | *Dice* |
|---|---|---|---|---|
| NUMSnet, Con | 68.22 | **79.1** | **57.08** | **59.42** |
| NUMSnet, GGO | 85.1 | **91.86** | **80.31** | **83.0** |
| NUMSnet, Lung | **99.36** | 93.29 | 92.76 | 95.22 |
| Unet, Con | 64.2 | 49.93 | 31.2 | 31.28 |
| Unet, GGO | 92.33 | 79.11 | 75.06 | 77.99 |
| Unet, Lung | 98.52 | 93.75 | 92.41 | 95.11 |
| wUnet, Con | **83.31** | 47.68 | 42.18 | 46.26 |
| wUnet, GGO | 89.41 | 86.61 | 79.4 | 81.99 |
| wUnet, Lung | 97.15 | **95.71** | **93.22** | **95.8** |
| Unet++, Con | 71.51 | 62.08 | 47.27 | 50.48 |
| Unet++, GGO | **94.14** | 71.92 | 69.83 | 73.03 |
| Unet++, Lung | 98.36 | 94.3 | 92.84 | 95.4 |

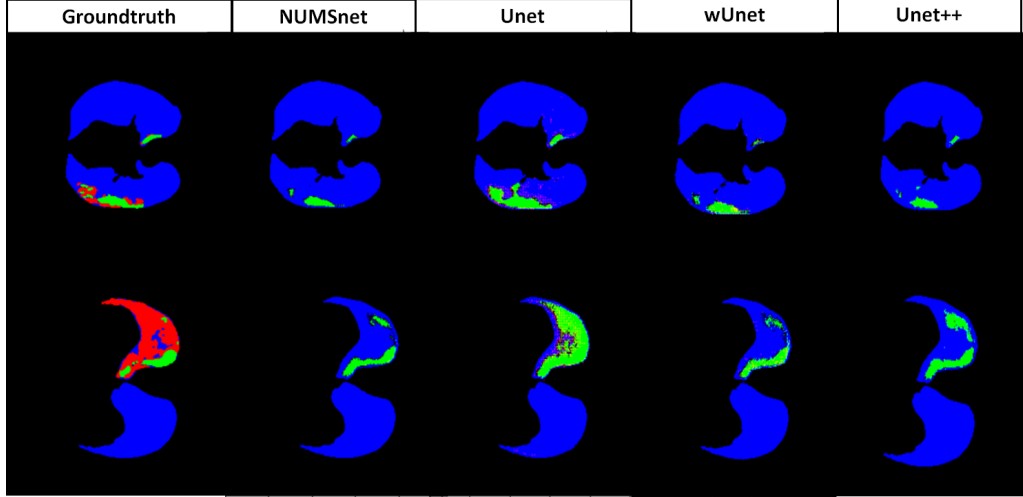

**Figure 7.** Example of Lung-CT segmentation by Unet variant models. **Row 1**: Best-case detection. **Row 2**: Worst-case detection. The color coding is as follows. Blue: lung regions; Red: GGO regions; Green: consolidation regions.

### 4.2. Sensitivity to Training Data

In this experiment, we modified the training dataset sequence and observed the segmentation performance variations. We comparatively analyze the performance for three sets of variations in training and test sequences. The first set comprised a training dataset that started with the first scan in the image stack as the reference image, followed by 10% of sequential images extracted per stack for training. All remaining images in the sequence were considered test samples, while 1% of the images from the test samples were withheld for hyper-parameterization as a validation dataset. This is called the [*Initial*, *Seq*] set. The second set comprised training images that started with the middle scan per 3D stack. Then, 10% of the subsequent scans were randomly selected while maintaining the order of images to generate the training sequence. All remaining images were used as test data, with 1% of the images randomly removed as the validation dataset. This is called the [*Mid*, *Rand*] set. The third set started with training images from the middle scan per stack, and 10% of the frames in a sequence were selected as training data. All remaining images were test data, with 1% of the images separated for validation tasks. This is called the [*Mid*, *Seq*] set. The variations in the multi-class semantic segmentation of the Lung-med and Lung-rad scans for all three training/test stacks are shown in Table 4.

**Table 4.** Comparative performance of NUMSnet on Lung-CT stacks when varying the training dataset, averaged over 5 runs. The best values for each metric are highlighted.

| Task | *Pr* | *Re* | *IoU* | *Dice* |
|---|---|---|---|---|
| Data: | Lung-med | | | |
| Initial, Seq, Con | **82.5** | 35.05 | 26.1 | 29.34 |
| Initial, Seq, GGO | **85.78** | 69.13 | 59.30 | 62.21 |
| Initial, Seq, Lung | 88.85 | **93.45** | 82.98 | 89.90 |
| Mid, Rand, Con | 60.38 | 96.52 | **57.97** | **57.97** |
| Mid, Rand, GGO | 70.15 | **99.46** | **69.75** | **69.75** |
| Mid Rand, Lung | **99.27** | 89.19 | **88.62** | **92.94** |
| Mid, Seq, Con | 60.37 | **97.32** | 58.91 | 58.91 |
| Mid, Seq, GGO | 70.15 | 93.17 | 68.01 | 68.01 |
| Mid, Seq, Lung | 98.73 | 89.28 | 88.27 | 92.59 |
| Data: | 10 Lung-rad | Stacks | | |
| Initial, Seq, Con | **87.74** | 44.24 | 38.94 | 43.13 |
| Initial, Seq, GGO | **92.23** | 75.69 | 72.46 | 75.19 |
| Initial, Seq, Lung | 95.44 | **96.74** | **92.87** | **95.79** |
| Mid, Rand, Con | 62.05 | **99.1** | **60.91** | **60.91** |
| Mid, Rand, GGO | 72.22 | **99.0** | 70.22 | 70.22 |
| Mid Rand, Lung | **99.79** | 91.74 | 91.6 | 94.57 |
| Mid, Seq, Con | 59.15 | 98.51 | 59.49 | 59.76 |
| Mid, Seq, GGO | 82.22 | 99.0 | **80.22** | **80.22** |
| Mid, Seq, Lung | 99.0 | 90.74 | 90.6 | 93.8 |

Here, we observe that the *IoU* and *Dice* scores for segmentation using the [*Initial*, *Seq*] training/test stack are consistently worse than those obtained using training sets that begin in the middle of each volume stack. This is intuitive since the initial layers often contain no annotations or minimal ROIs, being a precursor to the intended ROIs. Thus, using the [*Initial*, *Seq*] training dataset, the NUMSnet model does not learn enough to discern the small ROIs in this stack. We also observe that the performance of the [*Mid*, *Rand*] and [*Mid*, *Seq*] training stacks are similar to that of the Lung-med stack. In addition, we observe a 10% improvement in *Pr* and *D* for [*Mid*, *Seq*] over [*Mid*, *Rand*] for GGO segmentation only. Thus, selecting training images in the middle of 3D stacks with randomly ordered selection is important for training the multi-class NUMSnet model.

### 4.3. Performance Analysis for NUMSnet Variants

In the third experiment, we analyze the number of up-sampling layers that should be propagated to subsequent training scans for optimal multi-class segmentation tasks per volume. In Table 5, we report the segmentation performance of NUMS-all for Lung-CT stacks, where all 10 up-sampling layers are transmitted. Comparing the *Dice* scores for the Lung-med stack for NUMS-all with those of NUMSnet in Table 2, we observe that NUMS-all improves segmentation *Re* for the smaller ROIs of GGO and Con, but the overall segmentation performance across all ROIs remains comparable. We make similar observations for the 10 Lung-rad stacks when comparing Table 5 and Table 3. Thus, given that NUMS-all has higher computational complexity without a significant improvement in the overall segmentation performance, the NUMSnet model can be considered superior to NUMS-all while training with limited images.

**Table 5.** Performance of Lung-CT segmentation with NUMS-all model averaged across 5 runs.

| Data | Lung-Med | | | |
|---|---|---|---|---|
| **Task** | *Pr* | *Re* | *IoU* | *Dice* |
| NUMS-all, Con | 66.81 | 72.63 | 53.08 | 54.86 |
| NUMS-all, GGO | 83.11 | 91.06 | 78.09 | 81.02 |
| NUMS-all, Lung | 99.67 | 90.93 | 90.74 | 94.64 |
| Data | 10 Lung-rad | Stacks | | |
| NUMS-all, Con | 64.14 | 96.04 | 63.05 | 63.06 |
| NUMS-all, GGO | 86.97 | 92.34 | 81.82 | 84.34 |
| NUMS-all, Lung | 99.63 | 92.89 | 92.56 | 95.1 |

### 4.4. Transfer Learning for Heart-CT Images

In this experiment, we analyze the transfer learning capabilities of pre-trained Unet and variant models from the Lung-CT stack to the Heart-CT stack. The trained models from the Lung-med image stack were saved, all layers before the final layer were unfrozen, and the final layer dimensions were altered to be retrained on the Heart-CT dataset. The only difference in the Unet and variant models between the Lung-CT and the Heart-CT image sets is the final number of classes in the last layer X(1,5). Re-using the weights and biases of all other layers provides a warm start to the model and aids in faster convergence while training with randomly selected ordered images. For this experiment, the performance of each Unet variant in segmenting regions with the label pixel values [205, 420, 500, 550, 600, 820, 850] are represented by the model name and [$pix_{205}$, $pix_{420}$, $pix_{500}$, $pix_{550}$, $pix_{600}$, $pix_{820}$, $pix_{850}$], respectively, in Table 6. Here, we observe that the NUMSnet has superior segmentation performance for the smaller ROIs with the pixel values [500, 550, 600, 820, 850], respectively, with 2–16% improvements in *Dice* scores for these regions over the Unet++ model. Thus, the NUMSnet model aids in transfer learning across anatomical image stacks and across label types and yields higher precision when segmenting smaller ROIs.

**Table 6.** Averaged performance of the Unet and variant models on 10 Heart-CT stacks across 5 runs. The best values for each metric are highlighted.

| Task | *Pr* | *Re* | *IoU* | *Dice* |
|---|---|---|---|---|
| NUMSnet, $pix_{205}$ | **96.2** | 78.83 | 75.53 | 78.01 |
| NUMSnet, $pix_{420}$ | **96.89** | 86.2 | 83.42 | 85.04 |
| NUMSnet, $pix_{500}$ | 94.84 | **98.16** | **93.29** | **95** |
| NUMSnet, $pix_{550}$ | **96.61** | **86.23** | **83.4** | **85.8** |
| NUMSnet, $pix_{600}$ | 94.95 | 80.26 | **76.03** | 79.28 |
| NUMSnet, $pix_{820}$ | **98.42** | 96.84 | 95.55 | 96.88 |
| NUMSnet, $pix_{850}$ | **90.41** | 81.55 | **73.03** | **75.05** |

**Table 6.** *Cont.*

| Task | Pr | Re | IoU | Dice |
|---|---|---|---|---|
| Unet, pix$_{205}$ | 95.01 | **79.17** | 75.48 | **79.3** |
| Unet, pix$_{420}$ | 95.54 | 88.39 | 85.31 | **87.78** |
| Unet, pix$_{500}$ | 94.8 | 95.08 | 91.06 | 93.34 |
| Unet, pix$_{550}$ | 92.27 | 82.5 | 76.44 | 80.49 |
| Unet, pix$_{600}$ | 90.09 | **83.1** | 74.84 | 79.23 |
| Unet, pix$_{820}$ | 97.53 | 80.95 | 79.43 | 81.93 |
| Unet, pix$_{850}$ | 88.1 | 46.81 | 44.01 | 46.81 |
| wUnet, pix$_{205}$ | 95.93 | 75.18 | 72.37 | 76.02 |
| wUnet, pix$_{420}$ | 94.64 | **91.75** | 87.37 | 89.96 |
| wUnet, pix$_{500}$ | 94.42 | 95.05 | 90.73 | 93.11 |
| wUnet, pix$_{550}$ | 89.27 | 64.52 | 57.46 | 61.54 |
| wUnet, pix$_{600}$ | 90.93 | 80.84 | 72.93 | 77.22 |
| wUnet, pix$_{820}$ | 95.3 | 88.77 | 84.91 | 87.99 |
| wUnet, pix$_{850}$ | 84.37 | 69.65 | 60.09 | 62.74 |
| Unet++, pix$_{205}$ | 96.11 | 67.93 | 65.01 | 68.82 |
| Unet++, pix$_{420}$ | 94.69 | 88.92 | 84.59 | 86.91 |
| Unet++, pix$_{500}$ | **97.06** | 92.21 | 89.93 | 92.46 |
| Unet++, pix$_{550}$ | 88.46 | 73.42 | 63.44 | 67.36 |
| Unet++, pix$_{600}$ | 94.21 | 73.17 | 69.7 | 73.09 |
| Unet++, pix$_{820}$ | 96.07 | 88.15 | 85.06 | 86.96 |
| Unet++, pix$_{850}$ | 65.06 | **99.95** | 65.07 | 65.07 |

Some examples of good and average segmentation using the Unet model variants on the Heart-CT stack are shown in Figure 8. Here, we observe significant variations for smaller ROIs across the Unet model variants.

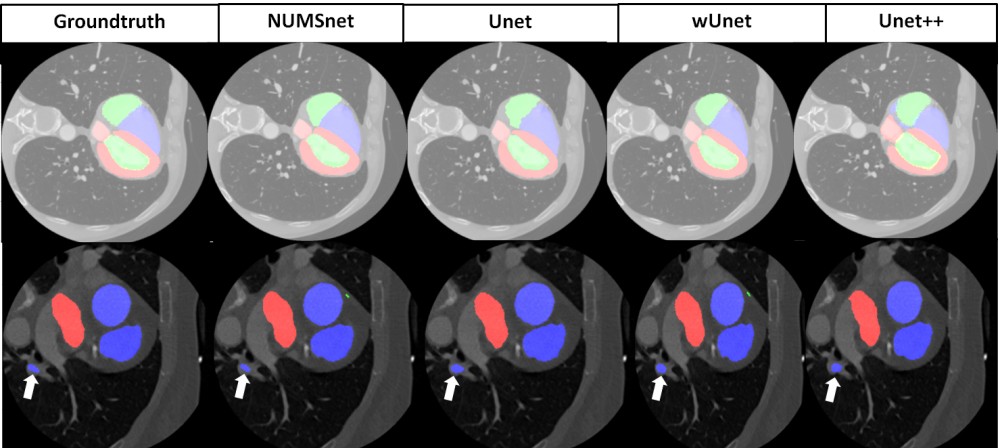

**Figure 8.** Examples of Heart-CT segmentation by the Unet variant models. **Row 1**: Good segmentation. **Row 2**: Average segmentation. In Row 2, we observe variations in the small ROI across Unet variants shown by the white arrow.

### 4.5. Ablation Study and Comparative Assessment

Finally, we analyze the importance of each hidden layer to the superior performance of the NUMSnet model when compared to the other Unet variant models. In this ablation study, we compared the performance of NUMSnet, with 11,713,943 total parameters, with its versions when only one hidden layer from the first level from the global feature layer (i.e., layer X(3,2) in Figure 3) is transmitted. This NUMSnet version is called NUMSnet$_{l1}$, and it has 11,349,628 parameters, of which 11,347,528 are trainable. Next, we generated a NUMSnet version in which the first two levels from the global feature level, i.e., layers (X(2,2), X(2,3) and X(3,2) from Figure 3), are transmitted. This is called NUMSnet$_{l12}$, and it has 11,614,508 parameters, of which 11,612,408 are trainable. The comparative

performance of NUMSnet$_{l1}$ and NUMSnet$_{l12}$ on the Lung-CT and Heart-CT datasets is shown in Table 7. Here, we observe that for the Lung-med stack, *Re* improves as more hidden layers are transmitted. However, for the shorter Lung-rad stacks, there is a small significant improvement in segmentation performance when increasing the number of transmitted layers from NUMSnet$_{l1}$ to NUMSnet$_{l12}$. We also observe that for the Heart-CT stacks, transmitting more hidden layers significantly enhances *Re* and the overall segmentation performance. Thus, by comparing NUMSnet$_{l1}$ and NUMSnet$_{l12}$ in Table 7 with the NUMSnet performance in Table 2, Table 3 and Table 6 for the Lung-CT and Heart-CT stacks, respectively, we conclude that the transmission of all six hidden layers in the NUMSnet model ensures superior segmentation performance across datasets.

**Table 7.** Comparative multi-class segmentation performance of the NUMSnet model variants.

| Data | Lung-Med | | | |
|---|---|---|---|---|
| Task | *Pr* | *Re* | *IoU* | *Dice* |
| NUMSnet$_{l1}$, Con | 70.48 | 71.34 | 51.14 | 53.1 |
| NUMSnet$_{l1}$, GGO | 91.11 | 80.6 | 75.62 | 78.48 |
| NUMSnet$_{l1}$, Lung | 99.61 | 90.55 | 90.24 | 93.99 |
| NUMSnet$_{l12}$, Con | 66.82 | 86.12 | 58.86 | 60.3 |
| NUMSnet$_{l12}$, GGO | 78.56 | 97.75 | 77.56 | 79.8 |
| NUMSnet$_{l12}$, Lung | 98.5 | 92.61 | 91.38 | 94.61 |
| Data | 10 Lung-rad | Stacks | | |
| NUMSnet$_{l1}$, Con | 67.66 | 75.28 | 51.86 | 53.67 |
| NUMSnet$_{l1}$, GGO | 89.67 | 88.22 | 80.63 | 83.38 |
| NUMSnet$_{l1}$, Lung | 99.51 | 93.16 | 92.72 | 95.2 |
| NUMSnet$_{l12}$, Con | 68.42 | 75.53 | 53.51 | 55.35 |
| NUMSnet$_{l12}$, GGO | 88.76 | 88.95 | 80.14 | 82.88 |
| NUMSnet$_{l12}$, Lung | 99.52 | 93.42 | 93.0 | 95.36 |
| Data | Heart-CT | | | |
| NUMSnet$_{l1}$, pix$_{205}$ | 97.38 | 67.21 | 65.51 | 68.85 |
| NUMSnet$_{l1}$, pix$_{420}$ | 94.92 | 88.71 | 84.42 | 86.03 |
| NUMSnet$_{l1}$, pix$_{500}$ | 97.29 | 94.56 | 92.34 | 94.31 |
| NUMSnet$_{l1}$, pix$_{550}$ | 90.03 | 87.87 | 78.95 | 82.43 |
| NUMSnet$_{l1}$, pix$_{600}$ | 90.9 | 75.95 | 68.3 | 71.9 |
| NUMSnet$_{l1}$, pix$_{820}$ | 97.91 | 87.58 | 85.82 | 87.43 |
| NUMSnet$_{l1}$, pix$_{850}$ | 89.96 | 75.67 | 68.12 | 70.17 |
| NUMSnet$_{l12}$, pix$_{205}$ | 96.63 | 78.18 | 75.65 | 78.36 |
| NUMSnet$_{l12}$, pix$_{420}$ | 95.9 | 83.1 | 80.24 | 81.93 |
| NUMSnet$_{l12}$, pix$_{500}$ | 97.44 | 90.48 | 88.44 | 90.33 |
| NUMSnet$_{l12}$, pix$_{550}$ | 94.49 | 78.9 | 74.98 | 77.97 |
| NUMSnet$_{l12}$, pix$_{600}$ | 93.76 | 76.93 | 71.15 | 74.16 |
| NUMSnet$_{l12}$, pix$_{820}$ | 98.52 | 90.13 | 88.84 | 90.19 |
| NUMSnet$_{l12}$, pix$_{850}$ | 89.07 | 86.02 | 76.42 | 78.34 |

Finally, the comparative performance of the NUMSnet model and previous works that trained deep learning models on larger training datasets is shown in Table 8. Here, we assessed the number of training images and training time on standalone GPU machines as an indicator of computational complexity, along with the segmentation performance or *Dice* scores for each output class category *i*. In addition, the previous works are identified as 2D vs. 3D based on the nature of convolutions in the implementations.

We observe that the proposed NUMSnet model achieves comparable or improved semantic segmentation performance across a variety of anatomical CT image stacks with only a fraction of the training images. This demonstrates the importance of nested layer transmission for enhanced boundary segmentation, especially for relatively small ROIs. For the Lung-CT stacks, the work by Voulodimos et al. [21] introduced a few-shot method

using a Unet backbone for GGO segmentation only, and while this method achieved high precision and accuracy, it had low recall and *Dice* scores. In addition, for the same dataset, the work by Saood et al. [13] used a small fraction of the images for training and achieved better binary segmentation performance than multi-class segmentation performance. It is noteworthy that no prior works have bench-marked the segmentation performance for the Lung-rad image stacks. For the Heart-CT stacks, most works have trained 3D Unet or 3D-segmentation models for voxel-level convolutions and trained on 20 CT stacks while testing on another 20 stacks for the high precision of segmentation per ROI. Our work is one of the few implementations of 2D convolutions on dense Heart-CT scans and the only work that evaluates the Heart-CT stacks in [30].

**Table 8.** Comparative performance of NUMSnet with respect to previous works.

| Method | Data | #Training Images | Metrics | Epochs/Training Time |
|---|---|---|---|---|
| Saood et al. (2D) [13] | Lung-med | 72 | $D_i = [22.5\text{–}60]\%$ | 160/25 min |
| Voulodimos (2D) [21] | Lung-med | 418 | $D_i = [65\text{–}85]\ \%(GGO)$ | ~210 s |
| Roychowdhury (2D) [35] | Lung-med | 40 | $D_i = 64\%\ (GGO)$ | 40/~70 s |
| NUMSnet (2D) | Lung-med | 82 | $D_i = [61\text{–}96\%]$ | 40/224 s |
| Payer et al. (3D) [22] | Heart-CT | 7831 | $D_i = [84\text{–}93\%]$ | 30,000/3–4 h |
| Wang et al. (3D) [15] | Heart-CT | 7831 | $D_i = [64.82\text{–}90.44\%]$ | 12,800/(Azure cloud) |
| Ye et al. (3D) [30] | Heart-CT | 7831 | $D_i = [86\text{–}96\%]$ | 60,000/~2–4 h |
| NUMSnet (Ours) (2D) | Heart-CT | 363 | $D_i = [75\text{–}97\%]$ | 60/362 s |

In Table 8, it is noteworthy that for Heart-CT segmentation, we applied a pre-trained model on Lung-CT and fine-tuned it on 4.6% of all hHeart-CT images to obtain similar segmentation performance. Additionally, we observe that the 3D-segmentation models yield stable *Dice* scores in a narrower range of 84–96% for Heart-CT data while taking several thousand epochs and several hours to train when compared to our work, which has a wider range of *Dice* scores but comparable performance for smaller ROIs to that in [15,22] and a training time of seconds. In addition, the work in [15] implemented 3D convolutions in a virtual machine in Azure cloud, so the training time is not comparable to those of standalone systems.

## 5. Discussion

The proposed NUMSnet model aims to reproduce the segmentation performance of 3D encoder–decoder models with 2D encoder–decoder equivalents, with the intention to scale the method across medical imaging modalities and scan densities. The current implementation is a NUMSnet model with a depth of 4, based on all the previous comparative works in Table 8, but the depth can be increased in future works based on the growing complexity, overlap and number of ROIs in medical image use cases. Additionally, the NUMSnet only uses 2D concatenations and convolution operations to ensure low additional computational complexity. However, for future works and specific complex use cases where the training time and computational complexity are not bottlenecks, some of the following three enhancements can be made. First, additional skip connections between scans can be added to combine up-sampled outcomes from previous scans to the current scans based on the underlying complexities of segmentation. Such skip connections will still have less training time and complexity when compared to 3D operations. Second, the encoder and decoder layers can be further enhanced with Resnet, Densenet and Retinanet backbones for segmentation enhancements in future works. Third, a combination of loss functions can be used at the block and scan levels for optimal parameterization.

Most existing deep learning models for multi-class semantic segmentation tasks have been developed at the image level to scale across imaging modalities [37]. However, for 3D medical image stacks, segmentation at a stack level minimizes irregularities related to varying imaging conditions, thereby resulting in superior region boundaries for small and

large ROIs. It is noteworthy that one key limiting condition for all semantic segmentation models is when the medical scans include written text on them. These irregularities can interfere with the segmentation of the outermost ROIs. In such situations, an overall mask can be generated and centered around all ROI regions and superimposed on the original image before passing it to the Unet and variant models, thus eliminating the written text region. Another alternative for reliable end-to-end segmentation in these cases, if enough annotated images are available, is to train two Unet or variant models to first detect the foreground region in the first Unet variant model, followed by segmenting the ROIs in the second Unet model, as shown in [15].

It is noteworthy that the single-stage Unet model and its variants are easily trainable with few annotated images, and they typically do not overfit. However, for high-resolution images, such as whole-slide images (WSI), where the dimensions of the medical images are a few thousand pixels per side, resizing such images to smaller dimensions to fit a Unet model or its variants may result in poor segmentation results [38]. In such scenarios, splitting the images into smaller patches, followed by training the Unet model and its variants, can improve the segmentation performance, as shown in [21].

A key consideration for multi-class segmentation using Unet variant models is the disparity between the ROI sizes, which can significantly impact the training stages when only a few annotated training images are available. For example, in the Lung-CT image stacks, the lung regions are larger than the GGO and consolidation areas, and because of this, using few training images and *Dice* coefficient loss over hundreds of epochs can bias the model to segment the lung region only. This occurs because the relative variation in pixel neighborhoods for larger ROIs is smaller than in pixel neighborhoods for smaller ROIs. In such situations, it is crucial to ensure that more training images are selected that have the smaller ROIs annotated and that the Unet variant models are run for about 40–60 epochs with region-sensitive loss functions.

Finally, for transfer learning applications, full image network weights transfer better when compared to Unet model variants trained on image patches, such as in [21]. This aligns with the works in [39,40], which demonstrate the ability of pre-trained models from one medical image modality to scale to other medical image stacks. Future efforts can be directed toward the transfer learning capabilities of the proposed NUMSnet model on WSI and patch image sets.

## 6. Conclusions

In this work, we present a novel NUMSnet model, which is a variation of the Unet++ model specifically for the multi-class semantic segmentation of 3D medical image stacks using only 10% of the images per stack, selected randomly in an ordered manner around the central scan of the 3D stacks. The novelty of this model lies in the temporal transmission of spatial pixel and neighborhood feature information across scans through nested layers. The proposed model enhances *Dice* scores over Unet++ and other Unet model variants by 2–9% in Lung-CT stacks and 2–16% in Heart-CT stacks. In addition, the NUMSnet is the only model that applies 2D convolutions for Heart-CT stack segmentation for the [30] dataset.

Additionally, in this work, we analyzed a variety of sampling methods to optimally select the minimal 10% training set. We conclude that the random selection of ordered scans is the optimal mechanism to select a minimal training set. Further, we analyzed the optimal number of up-sampling layers that should be transmitted for the best semantic segmentation performance. Here, we conclude that all six nested layers of a Unet++ model are significant for transmission, while adding additional up-sampling layers for transmission increases the overall computational complexity of the NUMSnet model while not significantly contributing to the segmentation performance for sparse training image sets.

Finally, we assessed the transfer learning capabilities of the NUMSnet model after it was pre-trained on Lung-CT stacks and fine-tuned on only 5% of available annotated Heart-CT images. We conclude that the NUMSnet model aids in transfer learning for similar medical image modalities, even if the number of classes and ROIs change significantly. Future work

can be directed toward extending the NUMSnet model to additional medical image modalities, such as X-rays, OCT, MRI stacks and RGB videos from dental to colonoscopy use cases.

**Funding:** This research received no external funding.

**Data Availability Statement:** Not applicable.

**Conflicts of Interest:** The authors declare no conflict of interest.

## Appendix A. Model Graphs

The proposed NUMSnet model layers and interconnections are shown in Figure A1. The layer interconnections from the NUMS-all model are shown in Figure A2.

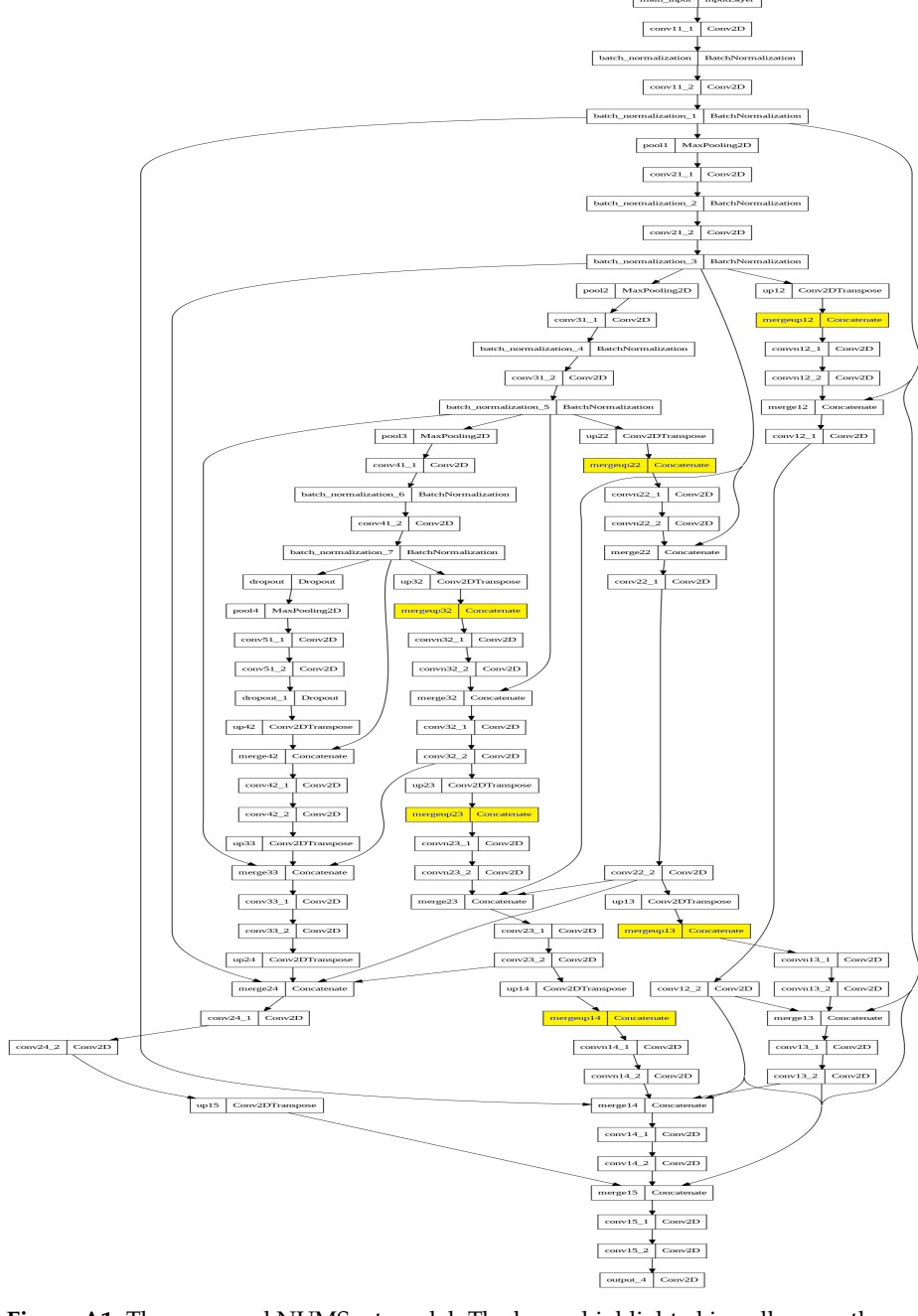

**Figure A1.** The proposed NUMSnet model. The layers highlighted in yellow are the new concatenation layers introduced by NUMSnet. All other layers are from the Unet++ model.

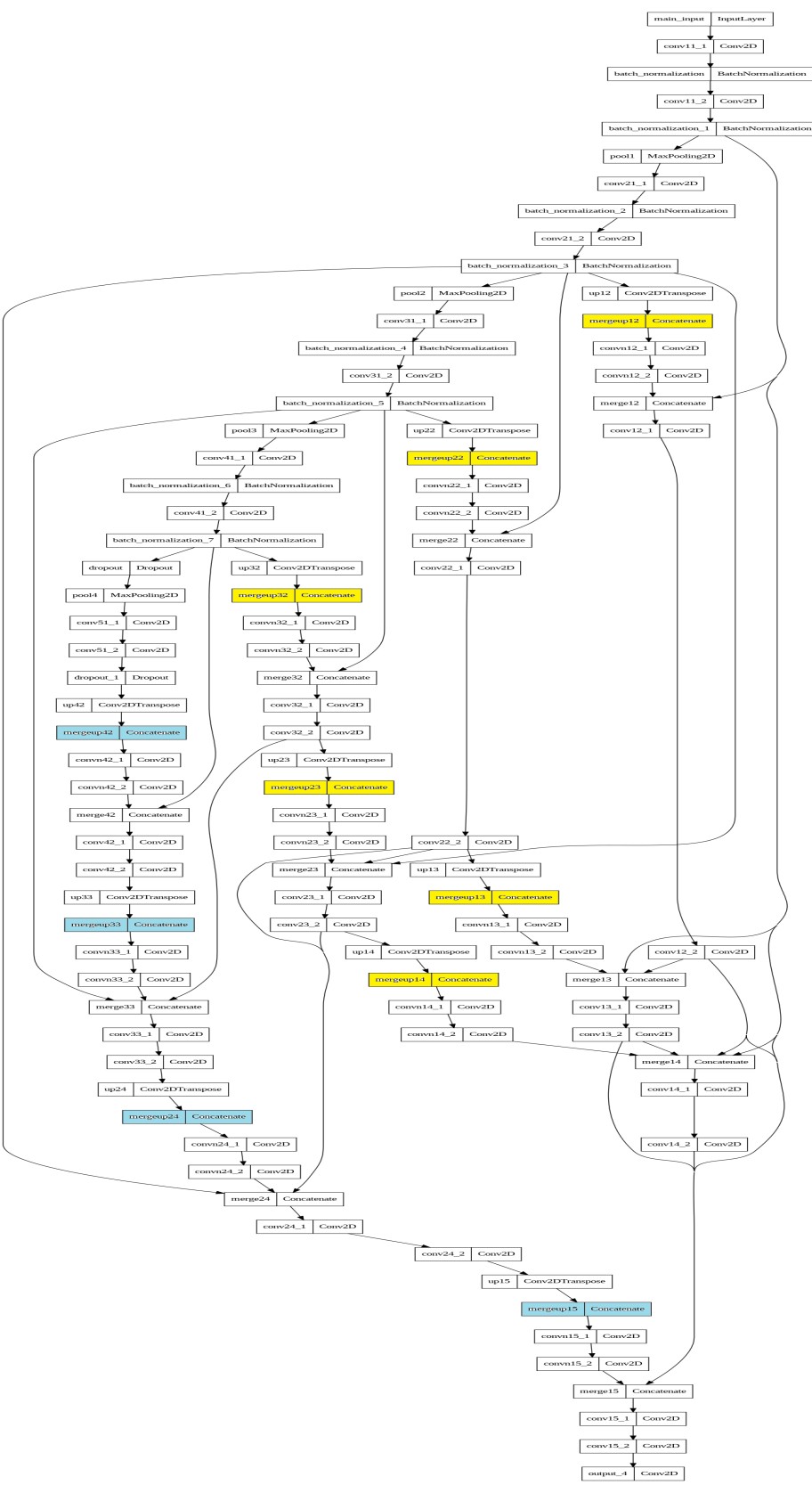

**Figure A2.** The NUMS-all model with all up-sampled layer features transmitted. The layers highlighted in yellow are the new NUMSnet concatenation layers. The layers highlighted in blue are the additional up-sampling layers in the NUMS-all model. All remaining layers are from the Unet++ model.

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
