# Peer review of "NUMSnet: Nested-U Multi-Class Segmentation Network for 3D Medical Image Stacks"

_information, doi:10.3390/info14060333_

Round 1
Reviewer 1 Report
Dear author,
I have read your article. I appreciate your effort. The paper is of good quality. Please address the changes and submit it back.
Abstract: Good
Introduction: You should start with deep learning and slowly move to medical imaging. You should also mention all the modalities such as ultrasound and x rays too. Also, include deep learning in medical imaging for various diseases. The following related articles will hopefully be of help.
1. Sampathila N, Chadaga K, Goswami N, Chadaga RP, Pandya M, Prabhu S, Bairy MG, Katta SS, Bhat D, Upadya SP. Customized Deep Learning Classifier for Detection of Acute Lymphoblastic Leukemia Using Blood Smear Images. InHealthcare 2022 Sep 20 (Vol. 10, No. 10, p. 1812). MDPI.
2. Chadaga K, Prabhu S, Sampathila N, Nireshwalya S, Katta SS, Tan RS, Acharya UR. Application of Artificial Intelligence Techniques for Monkeypox: A Systematic Review. Diagnostics. 2023 Feb 21;13(5):824.
All these use medical imaging techniques for diagnosis. You can add more articles too.
Related work: It is good , try to include other imaging modalites too.. You can include a table too to compare other studies.
3. Materials and Methods. It is written well. However, check for typos.
4. Results: It is good. .
Also , add challenges and future directions. Mention the limitations of your work.
Conclusion: Please remove citations from conclusion.
Discussion should be before conclusion! Further please make the discussion section more extensive.
I have no hestitation in accepting this manuscript once you address the changes. It is a good paper.
English is fine. Please check for typos. Proof read the matrix
Author Response
We would like to thank the reviewer for the invaluable comments. Based on all the reviewer comments, the manuscript has been revised thoroughly to reflect 4 major updates. First, we have modified the Introduction, Discussion and Conclusion sections to include more examples of Unet/segmentation algorithms for medical imaging domains. Second, we have added an ablation study to assess the importance of all the transmitted layers in the NUMSnet model in the experiments section. We create two variants of the NUMSnet model, namely NUMSnet_l1, where only one hidden layer at the first level after global feature layer is transmitted, and NUMSnet_l12, where 3 hidden layers from the first 2 levels after the global feature layers are transmitted. Comparing the NUMSnet_l1, NUMSnet_l12 , and NUMSnet models, we observe that all the 6 hidden layers are important for standardized multi-class semantic segmentations across imaging modalities. Third, we have rerun and updated all the Tables to reflect averaged performance metrics across 5 randomized runs and we have modified the Figures to ensure high resolution. Also, we have revised the manuscript to ensure accuracy and readability. Fourth, we have re-written the Data and Methods section to explain the mathematics at scan level and we compare the NUMSnet model with existing 2D and 3D segmentation variants. We have incorporated ALL the suggestions provided by the reviewer and we have added additional references and experiments to demonstrate the importance of Unet and variant models for other medical imaging domains. Responses to the comments are below.
Comment 1: Introduction: You should start with deep learning and slowly move to medical imaging. You should also mention all the modalities such as ultrasound and x rays too. Also, include deep learning in medical imaging for various diseases. The following related articles will hopefully be of help.
1. Sampathila N, Chadaga K, Goswami N, Chadaga RP, Pandya M, Prabhu S, Bairy MG, Katta SS, Bhat D, Upadya SP. Customized Deep Learning Classifier for Detection of Acute Lymphoblastic Leukemia Using Blood Smear Images. InHealthcare 2022 Sep 20 (Vol. 10, No. 10, p. 1812). MDPI.
2. Chadaga K, Prabhu S, Sampathila N, Nireshwalya S, Katta SS, Tan RS, Acharya UR. Application of Artificial Intelligence Techniques for Monkeypox: A Systematic Review. Diagnostics. 2023 Feb 21;13(5):824.
Ans: Many thanks for the suggestions. These references and several other have now been added to the updated manuscript.
Comment 2: Materials and Methods. It is written well. However, check for typos.
Ans. Many thanks for the comment. We have thoroughly checked the paper for typos and readability.
Comment 3: Add challenges and future directions. Mention the limitations of your work.
Ans: Many thanks for the comment. We have added challenges in the Discussion section. Future work has been added to the Conclusion section.
Comment 4: Conclusion: Please remove citations from conclusion.
Ans: Many thanks for the comment. This change has been made in the updated manuscript.
Comment 5: Discussion should be before conclusion! Further please make the discussion section more extensive.
Ans: Many thanks for the comment. This change has been made in the updated manuscript.
Comment 6: I have no hesitation in accepting this manuscript once you address the changes.
Ans: Many thanks for the comment. We highly appreciate your support for our work.
Reviewer 2 Report
This work presents a novel deep architecture based on UNet++ for the segmentation of 3D medical images. Experimental results shown that proposed NUMSnet outperforms several unet variants on Lung-CT and Heart-CT volumes. My comments are as follows.
Major issues:
1) The manuscript is not well organized. Section 3 is redundant, and the description about UNet++ should be simplified. Also, the authors could introduce the NUMSnet directly on section 3.2.
2) The main difference between NUMSnet and UNet++ is that NUMSnet incorporates the features from its previous scan. And, it can be observed that in Table 2 that NUMSnet outperforms UNet++ on three tasks. Indicating that transmitting cross-scan features is effective. I suggest adding ablation studies on the transmitted features to verify the role of high-level features and low-level features. In other words, is it necessary to transmit the features from all the 6 nested layers ? Are these 6 features all contribute to the performance improvement?
3) As far as I know, F1-score or Dice index = 2*Pr*Se/(Pr+Se). However, this equation does not hold on Tables 2, 3. Could the authors explain for this.
Minor issues:
1) Figures in this manuscript are with low quality. For example, Figure 4 and 5. It is better to adopt images with at least 300dpi.
2) Descriptions about running environment are better moved to section 4.
N/A
Author Response
We would like to thank the reviewer for the invaluable comments. Based on all the reviewer comments, the manuscript has been revised thoroughly to reflect 4 major updates. First, we have modified the Introduction, Discussion and Conclusion sections to include more examples of Unet/segmentation algorithms for medical imaging domains. Second, we have added an ablation study to assess the importance of all the transmitted layers in the NUMSnet model in the experiments section. We create two variants of the NUMSnet model, namely NUMSnet_l1, where only one hidden layer at the first level after global feature layer is transmitted, and NUMSnet_l12, where 3 hidden layers from the first 2 levels after the global feature layers are transmitted. Comparing the NUMSnet_l1, NUMSnet_l12 , and NUMSnet models, we observe that all the 6 hidden layers are important for standardized multi-class semantic segmentations across imaging modalities. Third, we have rerun and updated all the Tables to reflect averaged performance metrics across 5 randomized runs and we have modified the Figures to ensure high resolution. Also, we have revised the manuscript to ensure accuracy and readability. Fourth, we have re-written the Data and Methods section to explain the mathematics at scan level and we compare the NUMSnet model with existing 2D and 3D segmentation variants. Responses to the comments are below.
Comment 1: The manuscript is not well organized. Section 3 is redundant, and the description about UNet++ should be simplified. Also, the authors could introduce the NUMSnet directly on section 3.2.
Ans: Many thanks for the comment. Based on the suggestion here, Section 3 has been modified significantly. We have an Image Pre-processing section, followed by mathematical notations to describe the skip connections in Unet++ and the NUMSnet model.
Comment 2: The main difference between NUMSnet and UNet++ is that NUMSnet incorporates the features from its previous scan. And, it can be observed that in Table 2 that NUMSnet outperforms UNet++ on three tasks. Indicating that transmitting cross-scan features is effective. I suggest adding ablation studies on the transmitted features to verify the role of high-level features and low-level features. In other words, is it necessary to transmit the features from all the 6 nested layers ? Are these 6 features all contribute to the performance improvement?
Ans: Many thanks for this comment. Based on this comment we have added an ablation study subsection in Section 4 of the updated manuscript.
Comment 3: As far as I know, F1-score or Dice index = 2*Pr*Se/(Pr+Se). However, this equation does not hold on Tables 2, 3. Could the authors explain for this.
Ans: Many thanks for this comment! We have verified that the Dice scores per image is 2*Pr*Re/(Pr+Re) and we have added it to the equation (8). Table 2 and 3 have been updated with our rerun results. It is noteworthy that the Pre, Re and Dice scores in all the tables are averaged results from 5 runs and across images and hence the averaged Dice score will not be equal to the averaged Pre/Re combination. We have verified the same in all our Tables so many thanks for pointing this out.
Comment 4: Figures in this manuscript are with low quality. For example, Figure 4 and 5. It is better to adopt images with at least 300dpi.
Ans: Many Thanks for this comment. All the images have been enhanced to ensure higher image qualities in the updated manuscript.
Comment 5: Descriptions about running environment are better moved to section 4.
Ans: Many thanks for the comment. We have now moved all discussions on run times and complexities to the Section 4 of the updated manuscript.
Reviewer 3 Report
The author's proposed 3D version of the UNet++ model for multi-class segmentation of medical data achieves competitive performance on two datasets and demonstrates excellent robustness. By stacking and fusing nested layers in the scan dimensions before and after, the deep network captures more domain information, enhancing the performance of multi-class segmentation while reducing the demand for labeled data. However, the paper would benefit from further clarification:
1. The author should provide comparisons of complexity, \textbf{speed}, and accuracy with other UNet variant backbones.
2. As a 3D extension of the UNet model, the proposed method should compare with other 3D UNet models rather than just 2D UNet models.
3. The mathematical or visual descriptions of operations in the scan dimension, which is an important contribution to the paper, needs clearer explanation.
4. The author should provide more detailed explanations on how they choose the number and depth of nested layers and clarify the impact of these decisions on the model's performance.
5. On page 8, line 250, it is unclear why a 3x3 convolutional kernel changes [256x256x64] to [245x256x32]. This requires clarification.
Overall, this study presents promising results and demonstrates the potential of the proposed approach, but further clarifications and comparisons would strengthen the paper.
The English writing in the paper is good.
Author Response
We would like to thank the reviewer for the invaluable comments. Based on all the reviewer comments, the manuscript has been revised thoroughly to reflect 4 major updates. First, we have modified the Introduction, Discussion and Conclusion sections to include more examples of Unet/segmentation algorithms for medical imaging domains. Second, we have added an ablation study to assess the importance of all the transmitted layers in the NUMSnet model in the experiments section. We create two variants of the NUMSnet model, namely NUMSnet_l1, where only one hidden layer at the first level after global feature layer is transmitted, and NUMSnet_l12, where 3 hidden layers from the first 2 levels after the global feature layers are transmitted. Comparing the NUMSnet_l1, NUMSnet_l12 , and NUMSnet models, we observe that all the 6 hidden layers are important for standardized multi-class semantic segmentations across imaging modalities. Third, we have rerun and updated all the Tables to reflect averaged performance metrics across 5 randomized runs and we have modified the Figures to ensure high resolution. Also, we have revised the manuscript to ensure accuracy and readability. Fourth, we have re-written the Data and Methods section to explain the mathematics at scan level and we compare the NUMSnet model with existing 2D and 3D segmentation variants. Responses to the comments are below.
Comment 1. The author should provide comparisons of complexity, \textbf{speed}, and accuracy with other UNet variant backbones.
Ans: Many thanks for the comment. We have now added discussions regarding the train time and accuracy in Section 4 Table 8 of the updated manuscript. We have added additional references in the Section 2 Related works in the updated manuscript.
Comment 2. As a 3D extension of the UNet model, the proposed method should compare with other 3D UNet models rather than just 2D UNet models.
Ans: Many thanks for this comment. Based on this comment we have added the 2D and 3D models in the Related work section and we have compared performances to other 3D segmentation models in Section 4 subsection 5.
Comment 3. The mathematical or visual descriptions of operations in the scan dimension, which is an important contribution to the paper, needs clearer explanation.
Ans: Many thanks for this comment. Based on this comment we have now provided the mathematical insights into skip connections for the Unet++ and the NUMSnet models in equations 4 and 12 of the updated manuscript. The entire Section 2 of the paper has been re-written to provide insights into the encoder/decoder layers and their impact.
Comment 4. The author should provide more detailed explanations on how they choose the number and depth of nested layers and clarify the impact of these decisions on the model's performance.
Ans: Many thanks for this comment. We have provided clear explanations and references from existing works for the choice of Unet++ and NUMSnet of level 4. However, in the Section 5, Discussion, we have added details on how variations to the NUMs net can be incorporated based on the complexities of future use cases. We believe along with depth; the encoder decoder backbone architectures can also be varied based on the complexity of future use cases.
Comment 5. On page 8, line 250, it is unclear why a 3x3 convolutional kernel changes [256x256x64] to [245x256x32]. This requires clarification.
Ans: Many thanks for this comment. This typo has now been fixed in the updated manuscript and the manuscript has been thoroughly revised for typos.
Comment 6: Overall, this study presents promising results and demonstrates the potential of the proposed approach, but further clarifications and comparisons would strengthen the paper.
Ans: Many thanks for this comment. We have carefully revised the updated manuscript based on all the comments here and we highly appreciate your support in significantly improving the quality of the paper.
Round 2
Reviewer 2 Report
Many thanks for the revision. My concerns are well addressed.
No.